# Ocean temperature impact on ice shelf extent in the eastern Antarctic Peninsula

Johan Etourneau[1,2], Giovanni Sgubin[1], Xavier Crosta[1], Didier Swingedouw[1], Verónica Willmott[3,4], Loïc Barbara[5], Marie-Noëlle Houssais[6], Stefan Schouten[3,7], Jaap S. Sinninghe Damsté[3,7], Hugues Goosse [8], Carlota Escutia[2], Julien Crespin[1], Guillaume Massé[9] & Jung-Hyun Kim [3,10]

The recent thinning and retreat of Antarctic ice shelves has been attributed to both atmosphere and ocean warming. However, the lack of continuous, multi-year direct observations as well as limitations of climate and ice shelf models prevent a precise assessment on how the ocean forcing affects the fluctuations of a grounded and floating ice cap. Here we show that a +0.3–1.5 °C increase in subsurface ocean temperature (50–400 m) in the northeastern Antarctic Peninsula has driven to major collapse and recession of the regional ice shelf during both the instrumental period and the last 9000 years. Our projections following the representative concentration pathway 8.5 emission scenario from the Fifth Assessment Report of the Intergovernmental Panel on Climate Change reveal a +0.3 °C subsurface ocean temperature warming within the coming decades that will undoubtedly accelerate ice shelf melting, including the southernmost sector of the eastern Antarctic Peninsula.

[1] Instituto Andaluz de Ciencias de la Tierra, CSIC-Universidad Granada, 18100 Armilla (Granada), Spain. [2] UMR 5805 EPOC, EPHE/CNRS, Université de Bordeaux, 33615 Pessac, France. [3] NIOZ Royal Netherlands Institute for Sea Research, Utrecht University, 1790 Texel, The Netherlands. [4] International Cooperation Unit, Alfred Wegener Institute, 27570 Bremerhaven, Germany. [5] Instituto de Investigaciones Oceanologicas, Universidad Autónoma de Baja California, 22860 Baja California, Mexico. [6] UMR 7159 LOCEAN, CNRS/UPMC/MHNH/IRD, Université Pierre et Marie Curie, 75252 Paris, France. [7] Faculty of Geosciences, University of Utrecht, 3584 Utrecht, The Netherlands. [8] Earth and Life Institute, Université de Louvain, 1348 Louvain-la-Neuve, Belgium. [9] UMI CNRS Takuvik, Université de Laval, G1V0A6 Québec, QC, Canada. [10] Korea Polar Research Institute, 21990 Incheon, South Korea. Correspondence and requests for materials should be addressed to J.E. (email: johan.etourneau@iact.ugr-csic.es)

The Antarctic Peninsula has been one of the most rapidly warming regions of the world during the twentieth century[1] where ~75% of the ice shelves have already retreated over the past 50 years[2,3]. This retreat durably affects the stability of the regional glaciers and the ice sheet mass balance, which ultimately contributes to the eustatic sea-level rise. Specifically, in the eastern Antarctic Peninsula (EAP), the rapid warming observed since the 1970s[4] have had marked consequences on regional glaciers and ice shelves. One of the first major events occurred in 2002 when the Larsen B ice shelf in eastern Antarctic Peninsula (EAP) (Fig. 1) collapsed and lost an area of ~3250 km² by calving huge icebergs to the ocean[3]. A series of smaller—but significant—events occurred earlier in the northern part of the EAP with the collapse of the Larsen A and the Prince Gustav ice shelves in 1995, as well as that of the Larsen Inlet[3] in 1989 (Fig. 1). Since 2010, a giant crack has continuously incised the Larsen C ice shelf until it broke off in 2017 to form a massive iceberg of ~6000 km² (~9–12% of the total ice shelf)[5], thus drawing the premise of unprecedented major collapses in the near future.

Such successive events were initially hypothesized to have been mainly driven by Antarctic surface warming[6,7]. Indeed, a surface air temperature (SAT) increase of 2–3 °C has been observed[8] between the 1960s and the late 1990s (Fig. 2a), which could have directly impacted surface ice-melt[7], increased hydrofracturation[9], and indirectly glacier acceleration via enhanced precipitation[6]. However, an increase in the ocean heat content[2,6,10] can also substantially reduce the ice sheet extent through basal and frontal melting, ice shelves thinning and iceberg calving[6,11]. Although the oceanic impact remains difficult to quantify due to the lack of observational data, recent studies suggested that the effects of subsurface ocean warming could have contributed to more than 50% of the total ice loss in some Antarctica areas during the last few years, especially in the EAP region[2].

Indeed, the intrusion of relatively warm Circumpolar Deep Waters (CDW) onto the Antarctic shelf has been shown to promote ice shelf basal melting around Antarctica[2,6,10,11]. The warm deep water is upwelled across the continental shelf and channeled toward the grounding ice line through cavities and troughs[6,10,12–14]. In the Weddell Sea, the CDW enters as a coastal current from the east and circulates clockwise as Warm Deep Water (WDW) along the EAP shelf[15]. However, the recent and past activity of this warm water mass, including its temperature change and modification along its pathway to the continental shelf by mixing with shelf waters (see Supplementary Note 1, Supplementary Figure 1), are not well known, nor its impact on the ice shelf extent. In particular, variations in the intensity and position of the circumpolar Southern Westerly Winds (SWW) along the EAP may strongly modulate the upward transport of WDW onto the shelf through Ekman pumping.

Here we combine recent and geological records spanning the past 9000 years to investigate the variability of environmental conditions of the EAP on multiannual-to-millennial timescales and its effect on the ice shelf boundaries. Our results suggest that the ocean thermal forcing, tied to the circulation of the relatively warm WDW, has played a central role on the regional ice shelf instability during both the instrumental period and Holocene; a slight ocean warming contributing to substantial ice shelf collapse and regression. When extrapolating these results to the most pessimistic Intergovernmental Panel on Climate Change (IPCC) emission scenario, we show that a slight subsurface ocean warming will further accentuate the current erosion of the EAP ice shelf by the end of the twenty-first century.

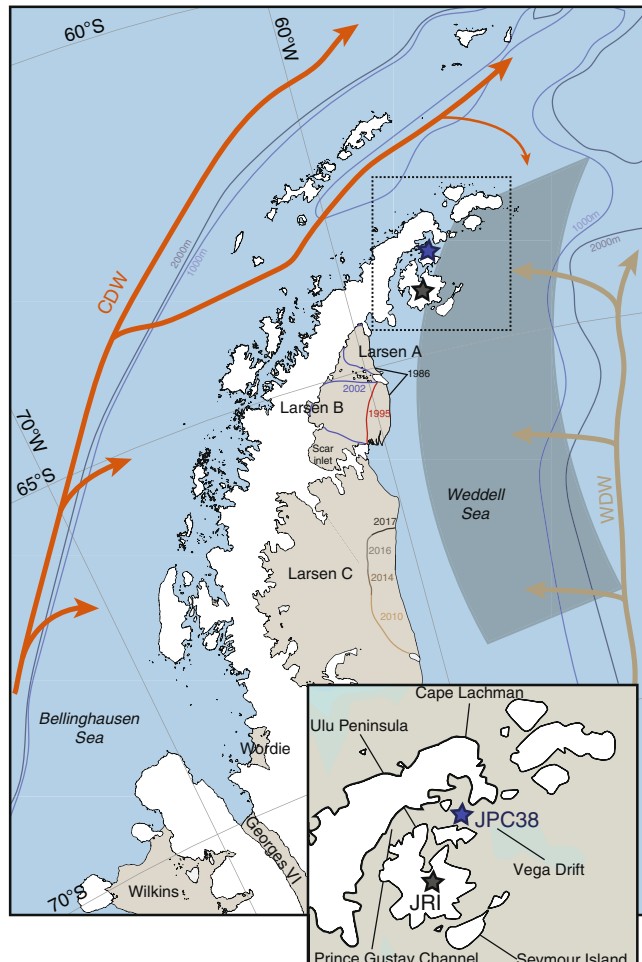

**Fig. 1** The Antarctic Peninsula and the recent shifts of the ice shelf boundary. Shown is the drastic retreat of ice shelves (Larsen A and B) along the EAP margin since 1986, the recent crack of the Larsen C since 2010, the study site (JPC-38) (63.717°S, 57.411°W, 760 m water depth) (blue star) and the JRI ice core (black star)[27]. The arrows show the circulation of the CDW (orange) and WDW (light brown) around the Peninsula. Reanalyzed data and model simulations have been computed along the EAP margin (i.e., in the gray area). Dark and light blue lines correspond to the bathymetry at 2000 and 1000 m depth and delineate the continental shelf from the abyssal Weddell Sea basin

## Results

**Ocean impact on recent ice shelf collapse**. Using reanalysis data[8,16], we first computed the variations of the Ekman pumping along the EAP margin since 1950 (Fig. 1) to decipher the link between wind forcing, in terms of strength and position, and evaluate its impact on the SOT. Ekman pumping anomalies experienced mainly negative values until the late 1970s and after the early 2000s (Fig. 2b), indicating a downward velocity. This dominant downwelling regime over our study area probably results from a northern position of the convergence zone between the SWW and the southerly winds, blowing northward along the south EAP (Supplementary Figure 2). Conversely, the anomalies switched toward positive values during the 1980s (Fig. 2b) to peak in the 1990s, thus implying a shift toward a dominant upwelling regime. This shift from regionally downwelling to upwelling conditions is probably tied to the intensification and poleward displacement of the SWW circulation at that time[17], which would have enlarged its influence at the expense of the northward wind

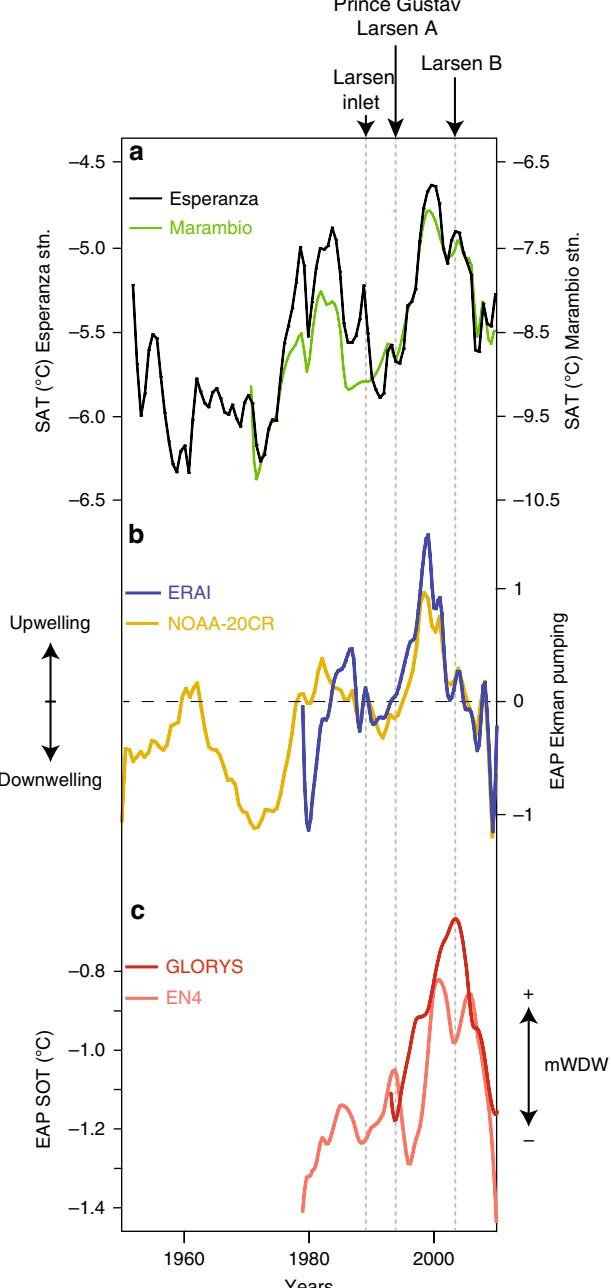

**Fig. 2** SAT, Ekman pumping, and SOT variations in the EAP during the instrumental period. **a** Monitored SAT (°C) at the Esperanza (black) and Marambio (green) stations since 1950 and 1970[25,26]. **b** Annual mean wind-forced Ekman pumping[16] (normalized series) along the EAP between 1950 and 2010 based on the reanalysis data from 20CR Project version 2[8] (orange) and ERA-interim[13] (blue). **c** Smoothed Annual mean SOT (°C) from 1993 to 2010 using GLORYS (dark red)[18] and between 1979 and 2010 using EN4 (light red)[19] along the EAP margin at 50–400 water depth. Black arrows indicate the major ice shelf collapse events. The smooth fit (also represented in Fig. 3) is an interpolate curve fit with a geometric weight applied using a Stineman function to the data. The weight is applied to 20% of the data (±10% of the data range, i.e., around the current point)

fields. As such, dominant SWW, offshore winds would have favored the upward transport of deeper water masses.

Reanalyses[18,19] concomitantly document a sharp increase in subsurface ocean temperatures (SOT) (50–400 m water depth) on the continental shelf reaching up to +0.6 °C, thus sharing a similar pattern with the computed Ekman pumping during the same period of time. Nevertheless, a certain time lag is sometimes observed between SOT and Ekman pumping variations, especially in the early 2000s. The subsurface temperature should quickly respond to variations in wind-driven upwelling of warmer deep ocean water. However, we suggest that the response is slowed down because of heat exchange between the upwelled water and the much colder atmosphere and surface waters. Other factors like a significant change in the temperature of the deep-water masses[20] may also have participated to the delay between the two records. We therefore argue that the SOT increase and associated warming of the EAP shelf resulted from two synergistic processes: (i) the enhanced penetration of warm deep waters on the shelf due to intensified wind-driven upwelling linked to a regional or global shift of the surface wind fields[21] (Fig. 2b), and (ii) the warming of Southern Ocean subsurface waters[20,22] and CDW during the last century[23,24].

We also find that the most rapid and abrupt SOT warming reconstructed around the EAP ice shelf over the last centuries concurred with the Larsen B ice shelf collapse, thus strongly suggesting that a slight SOT increase, in concert with atmospheric warming, has been pivotal in controlling the Larsen B ice shelf instability. While we miss robust observation data prior the 1970s, the continuous SOT increase during phases of smaller collapses may imply that SOTs have potentially contributed to other smaller ice shelf collapses along the EAP and around Antarctica. Comparatively, cold SOTs as observed in the early 2000s, probably due to a dominant downwelling forcing, during a period of pronounced SAT cooling[25,26], likely prevented further large ice shelf to collapse. Our results suggest that both the ocean (through wind-driven circulation and off-shelf water temperature changes) and atmosphere (through SAT and wind stress changes) forcing have closely acted together over the last decades to either strongly destabilize the EAP ice shelves or, inversely, to limit their dislocation.

**Ocean-driven ice shelf regression throughout the Holocene.** Beyond the observational period, there is only one available regional SAT record in the EAP[27] (Fig. 3a), situated in James Ross Island (JRI), and no SOT records so far. Based on water stable isotopes calibrated to recent air temperatures[7,27], the reconstructed mean annual SAT documents a 1.5 °C cooling over the Holocene occurring in two steps between 10,000 and 6000 years before present (BP), and 3500 and 500 years BP. The Holocene cooling was interrupted by a slightly warmer period. The first main cooling episode corresponds to a phase of major EAP ice shelf retreat reported in the literature[28–31]. Indeed, marine sedimentological data indicate that the northern JRI ice shelf initiated a transition from grounded to floating ice ~8000 years BP ago[30], which coincided with the early deglaciation of the Prince Gustav Channel and the Ulu Peninsula ice shelves, northwest of JRI[29]. This deglaciation preceded an ice free-regime that characterized the Vega drift and surrounding areas from ~7000 to 5000 years BP[30,32–34]. The Larsen A ice shelf was probably destabilized at least as early as ~6300 years BP[35], while evidence show that the Larsen B ice shelf experienced a continuous and significant shrinkage throughout the Holocene[28]. Hence, the EAP ice shelves underwent a major retreat mostly between ~8000 and 6000 years BP.

While the ice core-derived SAT were overall warmer throughout the Holocene than during the last two millennia and could have hence favored the EAP ice shelf surface melting during the entire period, the slow and gradual atmospheric cooling trend can hardly explain the initiation of the EAP ice shelf regression and its rapid disintegration between 8200 and 6000 years BP. Therefore,

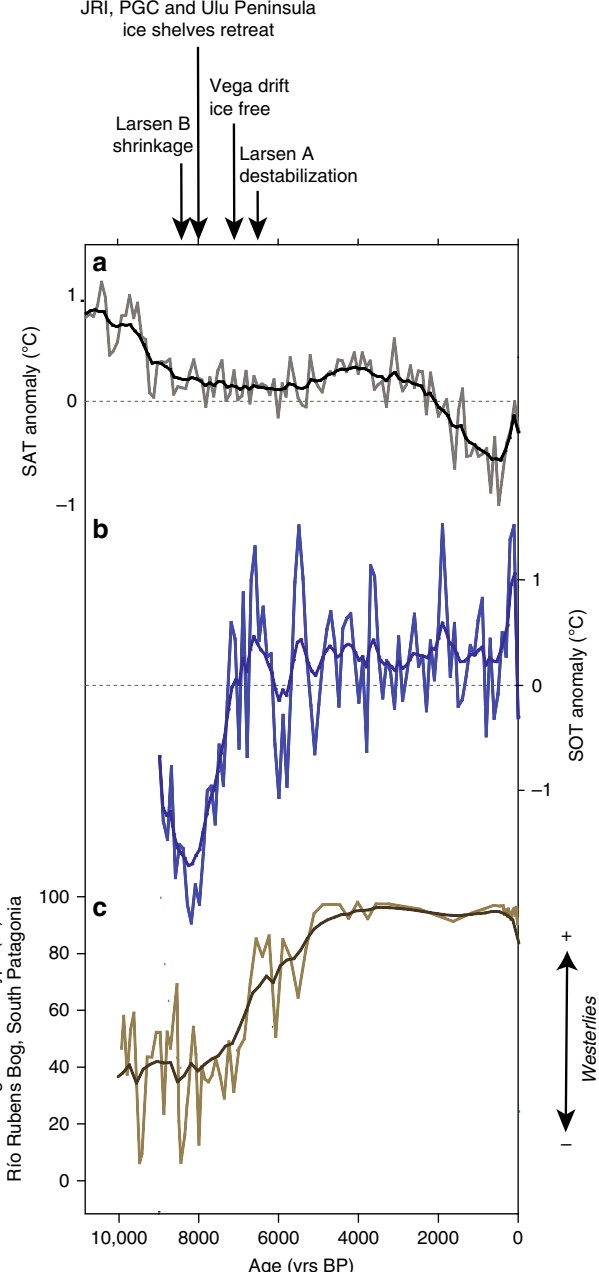

**Fig. 3** Holocene SAT, SOT, and SWW records. **a** The 100-year average SAT anomaly at the JRI core site (gray)[27], **b** the 100-year average TEX$_{86}^{L}$-derived SOT anomalies reconstructed at the JPC-38 (blue) and their respective smoothed records spanning the last 9000 years. **c** *Nothofagus* pollen record (%) at the Río Rubens Bog site, Patagonia, east of the Andes, Argentina, reflecting intensity and southward migration of the Westerlies[41–43]. The black arrows represent the major steps of the EAP ice shelf history[29–35]

other mechanisms must be investigated. A good candidate is an ocean warming at depth over the Holocene, as observed for the most recent times. To investigate such a potential link, we provide the first EAP SOT record spanning the last 9000 years at a 100-year resolution. We use the well-dated marine 20-m-long Jumbo Piston core NBP99–03–38 (JPC-38)[36] drilled within the Vega Drift in the northern Prince Gustav Channel (Fig. 1). The stratigraphy[36] is based on recent sediment using multi-core sections and strategic [14]C ages situated at each major SOT shifts, thus giving us confidence on the timing of the main changes.

Being only located 56 km north of the JRI ice core, it offers a unique opportunity to compare both oceanic and atmospheric evolutions at secular-to-millennial timescales and therefore to identify the primary forcing controlling the regional ice shelf dynamics. We apply the TEX$_{86}^{L}$ proxy (TetraEther Index of tetraethers with 86 carbons) for low temperatures that reflect mean annual SOT[37–39]. We mostly discuss here the TEX$_{86}^{L}$-converted temperature record in terms of trends and amplitudes rather than absolute values (see Methods).

Our record reveals a sharp SOT increase from ~8200 to 7000 years BP, followed by a continuous and slow ocean warming of +0.3 °C over the course of the mid-to-late Holocene (Fig. 3b), which is supported by diatom census counts in the same core[36]. A similar warming has been recorded over the nearby South Orkney Plateau, in the northeast of the Powell Basin, where diatom assemblages reveal a pronounced decline in winter sea-ice cover possibly associated with a warming ocean between ~8200 and 4800 years BP[40]. In contrast to the SAT, which started to cool ~10,000 years BP ago, the abrupt ocean warming recorded at the JPC-38 core site occurred synchronously with the inferred retreat of the EAP ice shelves. Several studies reported an intensification and southward migration of SWW during the same period of time[41–43], as illustrated for instance by pollen records in Patagonia[41], South America (Fig. 3c). Building on modern observations, we suggest that this reorganization in the wind fields, including the strengthening and poleward displacement of the convergence zone between the SWW and the northward winds, promoted the upwelling and intrusion of warm deep waters on the EAP continental shelf. We propose that the increase in SOT initiated the ice shelf regression in the northeast EAP through enhanced basal/frontal melting and possibly iceberg calving. Our assumption is consistent with some recent findings in the Amundsen Sea, in southwestern Antarctic Peninsula, where the ice shelf shrinkage during the early Holocene has been similarly attributed to CDW shoaling driven by a more southern position of the SWW during the early Holocene[44].

The long-term SOT increasing trend at the JPC-38 core site was punctuated by up to 1.5 °C warm events at the centennial scale, while cold events dwarfed over the course of the Holocene (Fig. 3b). Most of the gradual Holocene EAP ice shelf thinning and shrinking may be explained by the long-term subsurface ocean warming inferred from the TEX$_{86}^{L}$ record in core JPC-38, likely driven by enhanced warm deep water penetration towards the ice shelf. Moreover, episodic peaks in SOT throughout the Holocene, resulting from additional supply or warming (or both) of deep water, which may have prevented the ice shelf from re-expanding during colder atmospheric phases, thus causing its progressive disintegration.

Estimating the impact of such an ocean warming on the ice shelf basal melting throughout the Holocene is not straightforward as it depends on many complex processes. Nevertheless, an order of magnitude including large uncertainties can be obtained using a simple parameterization[45]. We assume a linear relationship between basal ice shelf melting and ocean temperature[45]. We consider an ice shelf front with a typical length of 250 km, i.e., the length of the Larsen A, B and Prince Gustav Channel ice shelf fronts before collapse (i.e., 1970s), which is shorter to the early-to-mid Holocene period given the large extension of the ice shelves in the northern part of the EAP at this time. As SOT constraints, we used a +0.3 °C (Holocene trend) and +1.5 °C (early Holocene) warming.

When considering an average SOT warming of +0.3 °C as observed in our records over the last 7400 years BP, we estimate a minimum net heat flux from the ocean to the ice shelf of ~3 × 10$^{11}$ J s$^{-1}$ and a melting rate of ~3 × 10$^{10}$ m$^{3}$ year$^{-1}$ (or ~1 mSv) (see Methods). This would correspond to a melting of at least

~10 m of ice per year on a surface covering the whole ice shelf front and penetrating 12 km below the ice shelves. If instead we take the surface of the 1970s EAP ice shelves for which we have a reasonable estimate (around 12,300 km² when summing up the Larsen A and B and the Prince Gustav Channel ice shelf area), this would correspond to an average melt rate of the ice shelf of about 2.4 m per year. A +1.5 °C SOT warming as recorded between ~8200 and 7000 years BP in the EAP would imply a five times larger melt rate following the same parameterization. This clearly appears sufficient to have a large impact on ice shelf dynamics, especially as parameterizations using simple linear relations between ocean temperature and melting[45] likely provide a lower estimate[46]. Therefore, although both ocean and atmosphere thermal forcing impacts on ice shelf instability, we conclude that, during the ocean warming phases, the SOT must have played a major role on controlling the regional ice shelf retreat.

**Future projections**. In the light of these new findings, we scrutinize the evolution of the SAT and SOT in projections from 26 climate models in order to anticipate the possible evolution of the ice caps of the Antarctic Peninsula during the next century. We focus on two different IPCC scenarios (referred to as the RCP 2.6 and 8.5), differing by their radiative forcing trajectories throughout the twenty-first century[47] and representing two potential evolutions of the future greenhouse gas concentrations (Fig. 4a, b and Supplementary Figure 3). We averaged the results of the different model simulations, thus filtering out most of the model internal variability, which allows to retrieve the forced temperature signal considered as the main response to potential anthropogenic greenhouse gas emissions.

For the RCP2.6 scenario, we find that both SAT and SOT trends for the next century are relatively weak and remains in the range of reconstructed variations over the Holocene period, each of the trends being <+1 °C and +0.1 °C over the next 90 years, respectively (Fig. 4a). Placing these results in a Holocene context, regional temperatures variations suggest that the RCP2.6-forced simulated warming trends may lead to limited ice shelf disintegration in the near future, unless the ongoing warming may determine the ice shelf fate[48]. In comparison, for the RCP8.5 scenario, projections exhibit regional atmospheric and oceanic warming trends that are three-to-four times higher than in the RCP2.6 scenario (warming by up to +3.5–4.5 °C and +0.3–0.4 °C in the atmosphere and subsurface ocean by 2100, respectively) (Fig. 4b). Such subsurface ocean warming is comparable to the +0.3 °C increase inferred for the last 7400 years BP in core JPC-38 and similar to the SOT trend observed in reanalyses over the last decades.

Given the recent impact of the +0.3 °C warming on the ice shelf as well as during the Holocene, we anticipate an accelerated melting of EAP ice shelves over the next decades due to projected anthropogenic forcing. Together with the simulated +3 °C atmospheric warming, they will probably substantially increase the occurrence of huge collapses if the climatic trend continues to follow the RCP8.5 scenario. Since features such as wind-driven ocean circulation, sea-ice cover or the amount and distribution of the freshwater released to the ocean though ice-melt still need to be improved in climate models[49,50], the warming of ocean temperature may be subdued in model projections. Consequently, the impact of changes in the wind forcing[17] on the shoaling and further penetration of the warm deep waters on the continental shelf remains potentially underestimated. Thus, the future of the EAP ice shelf may be possibly more marked than projected. If the

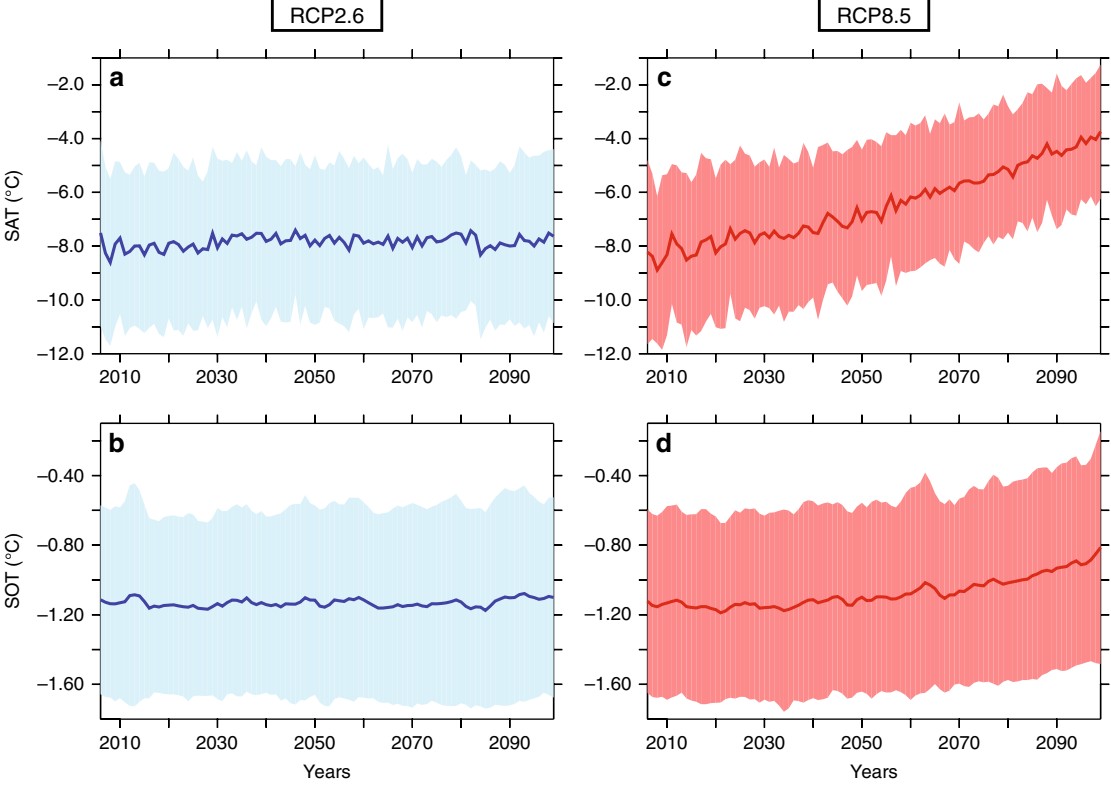

**Fig. 4** SAT and SOT projections from the present to 2100. Mean SAT and SOT (°C) simulations based on the **a**, **b** RCP2.6 and **c**, **d** RCP 8.5 scenarios[47] using 26 CMIP models along the EAP margin. The thick lines represent the ensemble mean. The overlaps correspond to one standard deviation of the ensemble

warm deep waters were to be redirected toward the continental shelf in the southernmost part of the Weddell Sea, enhanced heat content could increase by ~20% the yearly basal ice-melt underneath the Filchner-Ronne ice shelf[51]. Would such a predicted warming affect the whole Weddell Sea region, its impact on the surrounding ice sheets and glaciers would be marked for sea-level rise.

## Methods

**Reanalysis data.** The quality of reanalysis is relatively low before 1979 in the high latitudes of the Southern Hemisphere as few data were assimilated in this region due to the lack of observations. There is no observation data for the sub-superficial waters covering the last decades and century. We acknowledge that the reanalyses data that we use to reconstruct the wind-driven forcing on the SOT in this study are therefore relatively uncertain but should be robust enough to mirror the penetration of the WDW at least over the last decades[52]. The Ekman pumping has been computed along the EAP continental shelf using surface wind velocities coming from the recent twentieth century reanalysis NOAA (20CR) Project version 2[8], consisting of an ensemble of 56 realizations with 2° x 2° gridded 6-hourly weather data from 1950 to 2010, and from the ERA-Interim version 2.0[16] reanalyses (see code in Supplementary file 1). The definition of the Ekman pumping used is:

$$w_e = \text{curl}\left(\frac{\tau}{\rho_0 f}\right)$$

where $\tau$ is the wind stress vector, $\rho_0$ is a reference density of freshwater, $f$ is the Coriolis parameter. The unit of Ekman pumping $w_e$ is m s$^{-1}$ but here it has been standardized by the standard deviation computed over the whole period of available data.

The SOTs were estimated from both the GLORYS reanalysis and EN4 objective reanalysis, each covering a different time interval. GLORYS produces monthly global ocean reanalysis at eddy-permitting resolution from 1993 to 2010[18], while EN4 generates monthly data that covers the period between 1979 and 2010[19]. GLORYS reanalysis is based on the ocean and sea-ice general circulation model NEMO in the ORCA025 configuration forced by surface boundary conditions derived from atmospheric ECMWF reanalysis, and on the assimilation of in situ temperature and salinity profiles observations as well as sea surface temperature from satellite measurements and sea-level anomalies obtained from satellite altimetry[18]. EN4 data set is an incremental development of the previous version EN3 and consists of an objective analysis based on the temperature and salinity profiles derived from WOD09, GTSPP, Argo, and the ASBO project[19]. SOT were estimated at a 50–400 m water depth range on the continental shelf, where the warm deep waters mix with the continental shelf waters before altering the grounded ice shelf stability (see Supplementary Note 1).

**The TEX$_{86}^L$ proxy.** To reconstruct subsurface ocean conditions in the coastal Antarctic regions, we applied the TEX$_{86}^L$ (TetraEther Index of tetraethers with 86 carbons) proxy for low temperature polar regions[37,53]. This proxy is based on the relative distribution of Thaumarchaeotal isoprenoid glycerol dibiphytanyl glycerol tetraethers (GDGTs) suitable for reconstructing SOT in the polar oceans[37–39]. The analysis of GDGTs was conducted at the Royal Netherlands Institute for Sea Research (NIOZ) as previously described[37–39]. To convert TEX$_{86}^L$ values into sea subsurface temperatures, we used the calibration considering the following equation:

$$\text{TEX}_{86}^L = \log\left(\frac{[GDGT - 2]}{[GDGT - 1] + [GDGT - 2] + [GDGT - 3]}\right) \quad (1)$$

$$\text{SOT} = 50.8 * +36.1 (r^2 = 0.87, n = 396, p < 0.0001) \quad (2)$$

The SOT anomalies have been computed by subtracting 1.84 °C to the SOT values, 1.84 corresponding to the mean of the estimated SOT values using Eqs. (1) and (2) for the entire Holocene. To estimate the impact of the SOTs on the ice shelf throughout our record and calculate the corresponding ice melting rate as shown hereafter, we compute a temperature change of +1.5 and +0.3 °C over the early Holocene (8200–7000 years BP) and the Holocene trend (7000–0 years BP), respectively, based on a linear regression.

Given the shallow depth of our study site (760 m water depth), we suggest that the downward transport is probably relatively fast and the GDGTs compounds are attached to particles, which are produced from the melting of sea ice or ice shelf, fecal pellets or marine snow. We exclude any terrestrial organic matter that could influence the TEX$_{86}^L$ as the BIT index (Supplementary Figure 4), based on the marine versus terrestrial GDGTs, exhibits very low values (<0.1)[54,55]. In the Antarctic Peninsula, the only two studies focusing on GDGT producers have reported that the Thaumarchaeota species mostly reside within the 50–200 m

upper ocean layer (Supplementary Figure 1) and reach their highest abundance during the late winter and early spring seasons[56,57]. Given the relatively shallow water depth of our study site and the seasonality signal generally smoothed out in the sediment, we assume that the TEX$_{86}^L$-derived temperatures predominantly reflect changes in mean annual temperature of the subsurface waters influenced by the relatively warm water intrusions when they are the most influential on the ice shelf instability. Alternative indices based on GDGTs (i.e., TEX86) and associated calibrations[37,53] exists as well for reconstructing SOT; however, there are either calibrated towards sea surface temperatures (i.e., 0 m water depth)[58,59] or are much less effective for polar oceans[37]. Indeed, application of the TEX86 global calibration reveals values higher than +10 °C on average, some reaching up to almost +16 °C (data not shown), which are unrealistically warm for this region. Although the TEX$_{86}^L$ is calibrated using ocean temperatures taken at 0–200 m water depth[37], we believe that the estimated SOTs represent the 50–400 m layer, i.e., the whole water mass that flows westward below the surface layer. Looking into last decade's ocean temperature evolution of the EAP in two reanalyses, we do not detect any major temperature change between these two water depths. Indeed, the EN4 SOTs display similar absolute values and inter-annual variability since 1979 (Supplementary Figure 5a). GLORYS SOTs also demonstrate a similar inter-annual pattern between the different depth ranges but present a ~0.6 to ~0.8 °C shift towards warmer temperatures at depth (Supplementary Figure 5b). The inter-annual variability seems slightly tamed in surface compared to deeper depth ranges. The main difference between the two reanalyses resides in that GLORYS integrates more layers than EN4 and therefore allows a better assessment of the temperatures through the water column. The lower surface temperatures observed in GLORYS are consistent with observations showing that the surface waters are strongly influenced by autumn-winter cooling and sea-ice formation and melting, associated with the cold ASW and WW (Supplementary Note 1). In contrast, the deeper waters are warmer owing to the influential circulation of the relatively warm modified WDW. As a result, it is even possible that the TEX$_{86}^L$ calibration based on the 0–200 m water depth range slightly underestimates our reconstructed SOT changes throughout the Holocene both in terms of absolute values and amplitude. We suggest that further work on the link between the living depth of Thaumarchaeota species, the related synthesis of GDGTs and the WDW must be conducted as the TEX86 proxy is increasingly used in the Antarctic and Subantarctic areas to reconstruct paleotemperatures, given the lack of the commonly used other proxies (the alkenones, the Mg/Ca on foraminifera for instance). In this way, refining the existing calibration is fundamental and beneficial for future investigations around Antarctica at any timescales.

**Estimated ice melting rate.** Sub-ice shelf melting should ideally be estimated using an ocean model coupled to an ice shelf model in order to represent the ocean circulation in the cavity beneath the shelf, the processes controlling the melting and the feedback between this melt and circulation. This is unfortunately impossible at the scales investigated here, first of all because the precise topography of the ice shelves is largely unknown. We thus rely on a very simple parameterization[45], used for instance in ice sheet modeling[60], assuming a basic linear relationship between ice shelf melting and SOT[45].

The net heat flux ($Q_{oi}^{net}$) from the ocean to the ice shelf has therefore been computed based on the following bulk formulation [45]:

$$Q_{oi}^{net} = \rho c_{pw} \gamma_t (\text{SOT} - T_f) * A_{eff}$$

where $\rho$ is the density of seawater (1000 kg m$^{-3}$); $c_{pw}$ is the specific heat capacity of seawater (4000 J kg$^{-1}$ C$^{-1}$); $\gamma_t$ is the exchange coefficient (10$^{-4}$ ms$^{-1}$); $T_f$ is the freezing point temperature at the ice shelf base; $A_{eff}$ is the effective area for melting.

The total net melting rate can be then estimated as follows:

$$\Delta m^{net} = Q^{net} / (\rho_i L_i) \Delta t$$

where $\rho_i$ is the density of the ice shelf (920 kg m$^{-3}$); $L_i$ is the latent heat of fusion (334,000 J kg$^{-1}$). We assume[45] that $A_{eff}$ can be estimated from the length of the ice shelf front before the major collapses (i.e., prior to 1970) and a constant cross-shelf length taken equal to 12 km.

Given that this parameterization tends to underestimate the calculated melting[46], this means that the values provided here, already large in terms of contribution, represents a lower bound. The Holocene warming and the simulated one by the end of the century are of the same order (+0.3 °C). Despite the need to improve such estimation, we strongly believe that the +10 m of ice melting computed for the Holocene could give a good overview for the near future if the current climatic trend continues to follow the RCP8.5 scenario. Moreover, although the Larsen C seems to not be primarily impacted by the summer SOT as recently evidenced[61], due to the limited impact of the wind forcing (i.e., the Westerlies), we cannot exclude that a combined effect associating its recent collapse and the continuous southward migration of the Westerlies would not promote the WDW shoaling in this area in the future and therefore amplify its disintegration.

**Model projections (present to 2100).** We investigate projections from 26 climate models (Fig. S3) participating in the fifth Coupled Model Intercomparison Project (CMIP5) following two different emission scenarios (RCPs)[47]: RCP2.6 and RCP8.5.

Our selection includes the models providing for both scenarios the physical parameters that we analyzed here, i.e., SAT and SOT. The models fulfilling this criteria are the following: bcc-csm1-1-m, bcc-csm1–1, BNU-ESM, CanESM2, CCSM4, CESM1-CAM5, CNRM-CM5, CSIRO-Mk3–6–0, FGOALS-g2, FGOALS-s2, FIO-ESM, GFDL-CM3, GFDL-ESM2G, GISS-E2-H, GISS-E2-R, HadGEM2-AO, HadGEM2-ES, IPSL-CM5A-LR, IPSL-CM5A-MR, MIROC5, MIROC-ESM, MPI-ESM-LR, MPI-ESM-MR, MRI-CGCM3, NorESM1-ME, NorESM1-M (Fig. 4 and S3).

**Code availability**. The code used to characterize the Ekman Pumping can be found as supplementary file.

## Data availability
All data produced by this study are available from the corresponding author (johan.etourneau@iact.ugr-csic.es).

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

## Acknowledgements

J.E. and C.E. are financially supported by the Spanish Ministerio de Economia y Competitividad (CTM2014–60451-C2–1-P) co-funded by the European Union through FEDER funds. J.-H.K. was supported by the grants funded by the Korea Polar Research Institute (KOPRI, NRF-2015M1A5A1037243 and PE19010). S.S. and J.S.S.D. are supported by the Netherlands Earth System Science Center funded by the Dutch Ministry of Education and Science (OCW). G.S. and D.S. were funded by the EMBRACE project (European Union's FP7, Grant Number: 282672). We also acknowledge funding from the French ANR CLIMICE, ERC ICEPROXY 203441, ESF PolarClimate, HOLOCLIP 625 and FP7 Past4Future as well as the Netherlands Organisation of Scientific Research (NWO) through a VICI grant to S.S. The HOLOCLIP Project, a joint research project of ESF PolarCLIMATE programme, is funded by national contributions from Italy, France, Germany, Spain, Netherlands, Belgium and the United Kingdom. The research leading to these results has also received support from the European Union's Seventh Framework programme (FP7/2007–2013) under Grant Agreement No. 243908, "Past4Future, Climate change – Learning from the past climate". This is HOLOCLIP Contribution No. 32. We also thank J. Giraudeau for his constructive suggestions on the manuscript.

## Author contributions

J.E. designed and coordinated the study and led the writing. V.W., J.-H.K., L.B., S.S., and J.S. S.D. were responsible for the GDGT data acquisition. G.S. and D.S. analyzed the observation-based reanalysis data and model simulations. H.G. estimated the ice shelf melting. J.E., J.-H.K., X.C., G.S., J.C., H.G., M.N-H., and D.S. were in charge of the majority of the data interpretation. All authors commented on the manuscript and contributed to the writing.

## Additional information

**Competing interests:** The authors declare no competing interests.

