## [Peer Review File · Nature Communications]

Reviewer #1 (Remarks to the Author):

Generally I found this analysis convincing as to the importance of oceanic forcing for EAP ice shelf retreat, and worthy of publication in Nature Communications. I am not familiar with the approach used to derive the SOT anomaly time series, so leave it to the other reviewers to evaluate this record. Some improvements are needed.

The key part of the paper is Figure 3. There are some unstated caveats here. First, the James Ross Island record (part a) is not directly air temperature but rather stable isotopes calibrated to recent air temperatures. More importantly this represents the annual mean NOT the summer temperatures that are directly relevant to ice shelf melting (The Abram et al. 2013 paper helps out a lot in this regard). Looking at the record itself and interpreting it as reflecting summer air temperatures says the recent decades of LIS disintegration are in fact cooler than conditions that prevailed from 11,000 to 2,000 years BP so surface melting conditions should have been highly favorable for ice shelf disintegration for that entire time period. What makes the 8,500-6,000 years BP special is the rapidly warming ocean as shown here.

Second, the westerly wind results are from South America and South Georgia and are not necessarily reflective of conditions in the northern Antarctic Peninsula. Please present a convincing case that a similar westerly wind behavior is manifested in your study region.

The discussion of Antarctic Peninsula warming is weak; add a synthesis of the many British Antarctic Survey contributions. An important perspective is the recent cooling and its impact on melting – see the following:

Turner, J., and Coauthors, 2016: Absence of 21st century warming on Antarctic Peninsula consistent with natural variability. *Nature*, 535, 411, doi:10.1038/nature18645.

Oliva, M., F. Navarro, F. Hrbáček, A. Hernández, D. Nývlt, P. Pereira, J. Ruiz-Fernández, and R. Trigo, 2017: Recent regional climate cooling on the Antarctic Peninsula and associated impacts on the cryosphere. *Science of The Total Environment*, 580, 210–223, doi:10.1016/j.scitotenv.2016.12.030.

Notice the steep decrease in GLORYS SOTC after the early 2000s in Figure 3 that could be reflective of this recent summer cooling.

Other issues:

Figure 2: The quality of the atmospheric and oceanic reanalyses prior to the modern satellite era (1979) when the observational availability was minimal is greatly overstated. Why use 20CR to give you SAT when you have JRI proxy temperatures? The validity of the Ekman pumping calculations prior 1979 seem highly questionable. Similarly, I don't believe the SOTC variations from EN4. Start by considering the following:

Schneider, D. P., & Fogt, R. L. (2018). Artifacts in century-length atmospheric and coupled reanalyses over Antarctica due to historical data availability. *Geophysical Research Letters*, 45, 964–973. <https://doi.org/10.1002/2017GL076226>.

Figure S4 is missing.

Line 241: Given the many climate model challenges, it is equally likely that the warming may be unrealistically amplified so a more tempered discussion is needed.

Reviewer #2 (Remarks to the Author):

Please see attached pdf

Reviewer #3 (Remarks to the Author):

Review of Etourneau et al.

Etourneau et al presented a Holocene subsurface (50-400m) ocean temperature (SOT) record from the climatically sensitive but data-poor eastern Antarctic Peninsula. The Holocene SOT record is reconstructed using a paleoclimate proxy known as TEX86L, which is based on the archaeal tetraether lipids. The TEX86L-SOT record shows an overall warming trend in the past 9000years, with

a 2degC warming step between ~8ka to 7ka during which ice shelf destabilization and shrinkage occurred. Interestingly, the Holocene SOT warming is in contrast to the cooling trend in surface air temperature (SAT) inferred from an ice core record drilled on James Ross Island (JRI). To explain their Holocene temperature data, Etourneau et al extracted from several existing reanalysis products the 1900-2010 mean annual SAT for the JRI site and the EAP margin (regional mean; area defined in Figure 1), surface wind velocities to calculate Ekman pumping anomalies, as well as mean annual SOT for the same time period. Based on these reanalysis data, the authors suggested that the timing of Larsen B collapse coincided with a warm peak in SOT, both of which were preceded (by a few years) by a shift in hydrographic regime to upwelling conditions. They proposed that the warming trend in both SAT and SOT are due to both an enhanced penetration of warm deep waters due to upwelling and warming in subsurface/deep waters. Assuming that these same processes operated on longer time scales, they probably contributed to the warming seen in the Holocene TEX86L-SOT record. Next, assuming a linear relationship between SOT and ice shelf melting, the authors argued that the Holocene SOT warming could have caused the melting of 10m (0.3degC) to 50m (1.5degC) of ice per year. Similar extent of ice melt could be expected for the future under the worst RCP scenario (8.5) since the projected warming is in the same range as the Holocene warming.

The manuscript is generally well-written, barring a few typos. The research topic is timely and the Holocene data are of high value as they are from a understudied area. The major claim of the paper is that slight increases in the subsurface (50-400m) ocean temperature in the EAP played a dominant role in the ice shelf instability at decadal and secular timescales. The claim is novel as ice shelf instability is usually attributed to both ocean and atmospheric forcing. However, as detailed below, such a claim is currently not fully supported by the data and analyses presented in this study:

First, all the SOT data analyses were extracted from 50-400m because the authors claimed that this is "where the warm deep waters mix with the continental shelf waters before altering the grounded ice shelf stability". However, it is not clear from the Supplementary Information what is the significance of this range of water depths or if they in fact correspond to any particular water mass. The authors claimed that the subsurface water mass at the study site is likely modified WDW, based on two T and S depth profiles from two stations, which I suspect are taken from the Southern Ocean Atlas (SOA; see attached gif file "EAP stations") and not Ocean Data View as claimed by the authors (note: ODV is a data visualization software not a database). In SOA, the data from the same locations as that in Figure S1 were measured in February 1975 and February 1933, respectively. I am therefore not convinced that the comparison of data from these stations can be used to determine the source/depth range of water mass as they reflect the condition on a single day in different decades. This is a very critical issue for the study as it affects all the data analyses and interpretation, i.e. the water depth range of interest is 50-400m. Were it any other depth range e.g. 200-400m, the result will likely differ fundamentally.

Second, more discussion is needed for the Holocene TEX86L record, which is one of the main contributions of this study. Given the proximity of the study site to land, is there any significant

overprint of terrigenous GDGTs on their TEX86L record? The authors also need to demonstrate that their result is insensitive to the choice of TEX86 index (TEX86, TEX86H or TEX86L). Why not use the more commonly applied TEX86 or TEX86H as this will aid comparison with other studies? Also important is to justify why it is reasonable to apply a 0-200m calibration to reconstruct the water temperatures at 50-400m. The temperature gradient at different water depths tend to differ, thus can result in different calibrations which in turn lead to different reconstructed absolute temperature values and also the amplitude of temperature change (e.g. using the SST calibration vs 0-200m). The choice of calibration can therefore affect the Holocene warming and the subsequent quantification of ice melt. Also, is there any evidence that the archaea at the study site mostly occur within this depth range? Can the authors propose some mechanisms by which the GDGTs below photic zone can be exported to the seafloor?

Third, given the strong seasonality in polar regions, is it possible that the GDGTs mainly reflect summer/winter instead of mean annual condition? Can seasonality explain the opposite trends in proxy-derived Holocene SA and SOT records, especially given the consistent trend in SAT and SOT derived from reanalysis products and climate models? If yes, then the main claim of the study that ocean plays a more dominant role than the atmosphere in Holocene ice shelf instability will no longer hold. Further, if proxies do record seasonal signal, how reasonable is it to apply mechanisms derived from mean annual reanalysis/observed data to explain the variations in the Holocene proxy data? The data analyses presented here are based on annual mean temperatures but I wonder if, summer or annual mean temperature, is more relevant for ice shelf instability as melting mostly occurs in summer?

Fourth, as it is, the link between SOT and ice-shelf instability is established based on the seemingly contemporaneous Larsen B collapse and SOT peak around year 2000. However, SOT was low during the collapse of other ice shelves, and the Ekman pumping anomalies hovered around 0 indicating neither upwelling nor downwelling was dominant. In other words, the mechanisms proposed by the authors to link SOT and ice-shelf instability do not seem to apply to most past ice shelf collapse events. Therefore, what is the likelihood of these processes occurring during the Holocene and in the future? And importantly, are they more likely than the processes that govern the collapse of all the other events?

Fifth, the paper needs more detailed description of the analyses performed on reanalysis data and climate models to help others replicate their work. For instance, how were the 0.3degC and 1.5degC Holocene warming calculated? For the SOT from the GLORYS and EN4: what is the definition of EAP margin - is it the same as the grey box in Figure 1? Why is it better to examine the SOT from the EAP margin instead of the SOT at the JPC38 site even though the latter will allow a better comparison with the Holocene TEX86L record? Calculation steps for the ice shelf melt rate need to be outlined so that others can reproduce the result. Please find more suggestions below in the form of specific comments.

Specific comments

Line 42: Typo; "...Prince Gustave..." should be "...Prince Gustav..." Please correct this throughout the manuscript.

Line 80: "...from globally downwelling to upwelling conditions..." globally or regionally? The calculations are based on data from the EAP.

Line 87-90: I am not convinced by this argument. Stronger upwelling will bring more warm deep waters to intermediate depths, thus should increase the SOT within days - not years. On the other hand, the variability in SOT is much higher than that of SAT (Figure 2) in spite of a higher heat capacity in the ocean, whereas the SAT measured at Esperanza and Marambio stations show more variability which is more consistent with the SOT variability and Ekman pumping anomalies, including the peak at around year 2000. These rather curious results need to be further discussed.

Line 97-98: Not clear what is meant by "...a slight SOT increase in the context of atmospheric warming...". Please clarify.

Line 128-131: TEX86L and TEX86H are based on different indices, thus can potentially result in different reconstructed temperature variations, including its trend (e.g. Taylor et al 2013) and variability (e.g. highly variable TEX86 record at ODP1098 of Shevenell et al 2010 vs. smoother TEX86L record at JPC-10 by Etourneau et al 2013). It is therefore important to demonstrate (at least in the Supplementary Information) that the reconstructed temporal trend at the study site is insensitive to the choice of the proxy index. Furthermore, the amplitude of reconstructed temperature change relies on the calibration used (due to the temperature range in the regression). Therefore, there needs to be justifications for using the 0-200m calibration to reconstruct temperature at 50-400m because (1) the temperature gradient at 50-400m (in a spatial temperature calibration dataset) is different than that at 0-200m, using a 0-200m calibration will therefore lead to erroneous temperature estimates including the amplitude of change; (2) any evidence that the sedimentary TEX86L at the study site originates from the water depths of 50-400m? In the calibration study of Kim et al (2012) it is stated that "...TEX86L predominantly reflects subsurface (the depth interval of ~45-200m)...". Also, what are the mechanisms that can transport the lipids from subsurface ocean below 200m to the seafloor? (3) How does the TEX86L-inferred for the core-top compare to modern-day/late Holocene SOT?

Line 166: "constrains" should be "constraints".

Line 167-169: Unclear what are errors of a factor 2. Please rephrase.

Line 179-185: This argument is flawed. The calculation of the ice shelf melting is based on the assumption that SOT warming causes ice melting, therefore the result, ie. the calculated melt rate, cannot be presented as evidence for a dominant ocean forcing on regional ice shelf instability.

Line 193-195: Please show the simulation results from all the models in Supplementary Information as to rule out the possibility that the averaged trend is in fact dominated by a few models with relatively strong trends.

Line 210: Typo in "...due ton anthropogenic forcing..."; "ton" should be "to". I would add "projected" before "anthropogenic forcing" as those IPCC RCP scenarios are projections not reality.

Line 213: "though" should be "through".

Line 228-231: Given what's written here, I urge the authors to include the uncertainty of the ensemble mean (e.g. in the form of quantile or standard deviation) in Figure 2 so that the reader can judge how reliable the reanalysis data are. In addition it will also be useful to add a Supplementary Figure showing all the realizations.

Line 232: "...to face SAT with Ekman pumping and SOT data." Unclear sentence, please rephrase.

Line 233-235: More details on the calculation of Ekman pumping is needed in order to improve transparency and enable reproducibility in future studies. At the very least, the definition and unit should be stated. I also encourage the authors to share the code used to perform the data analyses.

Line 237: The SOT were estimated from BOTH the GLORYS and EN4. Please rephrase to reflect this. I would also briefly mention why both datasets are needed for the SOT estimation (i.e. each covering different time intervals).

Line 244 & 247: On what basis is the "initial temperature" for each reanalysis dataset determined?

Line 247-249: It is not very clear in the Supplementary Information how/why 50-400m is assumed. As shown in Figure S1, the thermocline at both sites extends down to 75m whereas the halocline extends further down to 200m. While the salinity profile at both sites do converge around 400m, the temperature profiles never do (the red site never approaches -1degC which characterizes the water mass).

SI Line 8: "...flew..." should be "...flowed..."; "...Gustave..." should be "...Gustav..."

SI Line 15: "...near stations" should be "...nearby stations"

SI Line 16: Which previous studies? References needed here.

SI Line 44-45: Missing reference / further evidence for "...among the best datasets", why/ how are they the best?

SI Line 47-50: To support your statement of "... share a high similarity" please provide statistics (e.g. correlation) or at least plot the reanalysis data next to the monitoring data in your Fig S2 to allow a better visual comparison. Also needed is justifications for opting reanalysis data over observed data at nearby sites (how near? provide coordinates of these sites or include them in one of the maps), especially in light of the structural similarity between the observed SAT time series and the SOT and Ekman pumping anomalies time series.

SI Line 55-57: Robust enough or not depends on the context. Further clarification on how/why given the regional oceanography features it is reasonable to assume that the reanalysis data based on few observations are adequate to support the claims made by the authors.

Figure S1: The authors need to clearly state the source of the hydrographic data used. Ocean Data View is a data visualization software not a database. It is also not clear if the data shown are annual mean or of certain month. Also needed here is a discussion on how well-constrained the hydrographic data are in this region and if the uncertainty may fundamentally change the working hypothesis (mWDW occurs between 50-400m at their study site).

Figure S4 is missing.

Response to reviewers

We would like first of all to thank the three reviewers for their very helpful, positive review and comments on our manuscript. Here we address a response to the Reviewers 1 and 3 as the Reviewer 2 does not ask for any clarifications or modifications.

Reviewer 1:

Generally I found this analysis convincing as to the importance of oceanic forcing for EAP ice shelf retreat, and worthy of publication in Nature Communications. I am not familiar with the approach used to derive the SOT anomaly time series, so leave it to the other reviewers to evaluate this record. Some improvements are needed.

The key part of the paper is Figure 3. There are some unstated caveats here. First, the James Ross Island record (part a) is not directly air temperature but rather stable isotopes calibrated to recent air temperatures. More importantly this represents the annual mean NOT the summer temperatures that are directly relevant to ice shelf melting (The Abram et al. 2013 paper helps out a lot in this regard). Looking at the record itself and interpreting it as reflecting summer air temperatures says the recent decades of LIS disintegration are in fact cooler than conditions that prevailed from 11,000 to 2,000 years BP so surface melting conditions should have been highly favorable for ice shelf disintegration for that entire time period. What makes the 8,500-6,000 years BP special is the rapidly warming ocean as shown here.

Response 1:

We added in the text, according to reviewer's comments, the exact meaning of the JRI ice core record. We also added that "While the ice core-derived SAT were overall warmer throughout the Holocene than during the last two millennia and could have hence favored the EAP ice shelf surface melting during the entire period, the slow and gradual atmospheric cooling trend can hardly explain the initiation of the EAP ice shelf regression and its rapid disintegration between 8,200 and 6,000 years BP."

Second, the westerly wind results are from South America and South Georgia and are not necessarily reflective of conditions in the northern Antarctic Peninsula. Please present a convincing case that a similar westerly wind behavior is manifested in your study region.

Response 2:

The Southern Westerly Winds (SWW) corresponds to a wind belt flowing eastward through the Drake Passage, between South America and the Antarctic Peninsula as well as all around Antarctica (Fig. R1 (below) and Supplementary Fig. 3). Unfortunately, to our knowledge, there are no existing Holocene SWW records in the Peninsula comparable to the ones generated in South America or in the Scotia Sea because most of the SWW records are based on subfossil pollen assemblages in peat and lake sediment cores, geochemical or mineralogical proxies reconstructing regional rainfalls, which remains difficult to apply in a cold icy area such as in the Peninsula. In our study we cite studies, which showed similar records at different locations, thus giving reliable reconstructions of the wind belt in the Drake Passage over the last 9,000 years in terms of paleointensity and paleoposition. Although we are perfectly aware, based on a compilation (Kilian and Lamy, 2012) or a comparison (Saunders et al., 2018) of several SWW reconstructions that there are still some uncertainties in terms of timing regarding the changes in position and strengthening of the westerlies, we selected the closest, most reliable and highly cited records.

Moreover, the striking resemblance between our SOT and wind records as exhibited in Fig.3 fully supports our hypothesis.

Figure R1. Here is highlighted in red the SWW showing that both JPC-38 site and the sites mentioned in the South America and Scotia Sea are both influenced by the same wind belt [Data/Image] from Climate Reanalyzer (<http://cci-reanalyzer.org>), Climate Change Institute, University of Maine, USA).

The discussion of Antarctic Peninsula warming is weak; add a synthesis of the many British Antarctic Survey contributions. An important perspective is the recent cooling and its impact on melting – see the following:

Turner, J., and Coauthors, 2016: Absence of 21st century warming on Antarctic Peninsula consistent with natural variability. *Nature*, 535, 411, doi:10.1038/nature18645.

Oliva, M., F. Navarro, F. Hrbáček, A. Hernández, D. Nývlt, P. Pereira, J. Ruiz-Fernández, and R. Trigo, 2017: Recent regional climate cooling on the Antarctic Peninsula and associated impacts on the cryosphere. *Science of The Total Environment*, 580, 210–223, doi:10.1016/j.scitotenv.2016.12.030.

Notice the steep decrease in GLORYS SOTC after the early 2000s in Figure 3 that could be reflective of this recent summer cooling.

Response 3:

We agree with the reviewer that we did not focus so much on the most recent cooling interval. As pointed out by the reviewer himself, the key part of our manuscript is about the Holocene records. Moreover as shown in Abram et al., 2013, it appears that a clear multidecadal cyclicity occurs during the last century, which remains poorly understood, although some assumptions relate this variability to phenomenon like the Pacific Decadal Oscillation (Zitto et al., 2015; Mayewski et al., 2016). Explaining the last cooling event would imply to mention all the others for the last century, for which we do not have robust oceanic data. Nevertheless, as the reviewer suggests, we

now mention this point in the first part of our manuscript as we are probably the first to show a concomitant variation between the atmosphere and ocean temperatures over this period (lines 103-105).

Other issues:

Figure 2: The quality of the atmospheric and oceanic reanalyses prior to the modern satellite era (1979) when the observational availability was minimal is greatly overstated. Why use 20CR to give you SAT when you have JRI proxy temperatures? The validity of the Ekman pumping calculations prior 1979 seem highly questionable. Similarly, I don't believe the SOTC variations from EN4. Start by considering the following:
Schneider, D. P., & Fogt, R. L. (2018). Artifacts in century-length atmospheric and coupled reanalyses over Antarctica due to historical data availability. *Geophysical Research Letters*, 45,964–973. <https://doi.org/10.1002/2017GL076226>.

Response 4:

In line with the reviewer's suggestion, we removed in the Fig.2a the reanalysis SOT records to replace them by observation data at both Esperanza and Marambio stations. Given the the lack of observations in ocean temperatures before the satellite era, we decided to keep GLORYS and EN4-derived SOT records solely since 1979. All the records presented in the revised version do not alter our hypothesis and even reinforce it.

Figure S4 is missing.

Response 5:

There are now 4 figures in the supplementary.

Line 241: Given the many climate model challenges, it is equally likely that the warming may be unrealistically amplified so a more tempered discussion is needed.

Response 6:

We smoothed our discussion as suggested by the reviewer.

Reviewer 3:

Etourneau et al presented a Holocene subsurface (50-400m) ocean temperature (SOT) record from the climatically sensitive but data-poor eastern Antarctic Peninsula. The Holocene SOT record is reconstructed using a paleoclimate proxy known as TEX86L, which is based on the archaeal tetraether lipids. The TEX86L-SOT record shows an overall warming trend in the past 9000years, with a 2degC warming step between ~8ka to 7ka during which ice shelf destabilization and shrinkage occurred. Interestingly, the Holocene SOT warming is in contrast to the cooling trend in surface air temperature (SAT) inferred from an ice core record drilled on James Ross Island (JRI). To explain their Holocene temperature data, Etourneau et al extracted from several existing reanalysis products the 1900-2010 mean annual SAT for the JRI site and the EAP margin (regional mean; area defined in Figure 1), surface wind velocities to calculate Ekman pumping anomalies, as well as mean annual SOT for the same time period. Based on these reanalysis data, the authors suggested that the timing of Larsen B collapse coincided with a warm peak in SOT, both of which were preceded (by a few years) by a shift in hydrographic regime to upwelling conditions. They proposed that the warming trend in both SAT and SOT are due to both an enhanced penetration of warm deep waters due to upwelling and warming in subsurface/deep waters. Assuming that these same processes operated on longer time scales, they probably contributed to the warming seen in the Holocene TEX86L-SOT record. Next, assuming a linear relationship between SOT and ice shelf melting, the authors argued that the Holocene SOT warming could have caused the melting of 10m (0.3degC) to 50m (1.5degC) of ice per year. Similar extent of ice melt could be expected for

the future under the worst RCP scenario (8.5) since the projected warming is in the same range as the Holocene warming.

The manuscript is generally well-written, barring a few typos. The research topic is timely and the Holocene data are of high value as they are from a understudied area. The major claim of the paper is that slight increases in the subsurface (50-400m) ocean temperature in the EAP played a dominant role in the ice shelf instability at decadal and secular timescales. The claim is novel as ice shelf instability is usually attributed to both ocean and atmospheric forcing. However, as detailed below, such a claim is currently not fully supported by the data and analyses presented in this study:

First, all the SOT data analyses were extracted from 50-400m because the authors claimed that this is "where the warm deep waters mix with the continental shelf waters before altering the grounded ice shelf stability". However, it is not clear from the Supplementary Information what is the significance of this range of water depths or if they in fact correspond to any particular water mass. The authors claimed that the subsurface water mass at the study site is likely modified WDW, based on two T and S depth profiles from two stations, which I suspect are taken from the Southern Ocean Atlas (SOA; see attached gif file "EAP stations") and not Ocean Data View as claimed by the authors (note: ODV is a data visualization software not a database). In SOA, the data from the same locations as that in Figure S1 were measured in February 1975 and February 1933, respectively. I am therefore not convinced that the comparison of data from these stations can be used to determine the source/depth range of water mass as they reflect the condition on a single day in different decades. This is a very critical issue for the study as it affects all the data analyses and interpretation, i.e. the water depth range of interest is 50-400m. Were it any other depth range e.g. 200-400m, the result will likely differ fundamentally.

Response 7:

Unfortunately, there are no published temperature (T), salinity (S), or any isotopic data, allowing us to directly measure the water masses flowing through the Gustav Channel and, more particularly, determine the depth range and temperature of the mWDW inflow. We therefore crossed a series of evidence in order to assess them:

1. Carmerlenghi et al. 2001 described changes in current velocity, clearly tracing three different water masses, two flowing eastward (surface and bottom), and one intermediate (subsurface) flowing westward.

2. We therefore attempted to define what is this intermediate water mass flowing through the Gustav channel, given that the Traumarcheota, the GDGTs producers, mostly grow at this water depth. There are very little T and S profiles available. The few accessible are indeed coming from the Southern Ocean atlas (confusingly named in our manuscript from ODV (it has been corrected)) and collected in February. The profiles reveal the presence of relatively warm water masses with the same characteristic as the mWDW found elsewhere around Antarctica (e.g. Herraiz Borreguero et al., 2015).

3. To better constrain our assumptions, we looked at the very few existing observation data from van Caspen et al. (2015), who showed that the WDW was flowing along the EAP continental shelf, before being redirected towards the ice shelf. This is supported by Nicholls et al., 2012 and more recently by Scambos and Klinger from The Larissa Project (POLAR2018 meeting: Early Weakening of the Larsen B Ice Shelf Prior to Break-up (unpublished data)).

4. According to numerous studies conducted on the Antarctic continental shelf in different areas, we know that the mCDW (mWDW in the Weddell Sea), flows below the subsurface, down to 400m. Indeed, Herraiz Borreguero et al., 2015 observed a flow of mCDW in Prydz Bay at this intermediate depth. Jacobs et al. (1992) and Joughin and Padman (2003) proposed that the mCDW affect the ice shelf base at 200-400m water

depth, thus supporting numerous previous studies (Foldvik et al., 2004; Herraiz-Borregero et al.; 2015; Liu et al., 2017; Nitsche et al., 2017; Walker and Gardner, 2017).

5. Although we do not have a direct evidence, we deduced that the intermediate water mass entering the Prince Gustav Channel, based on all these evidence, is most likely related to the mWDW flow.

Regarding the influence of the 50-400m vs 200-400m water depth, we do not find any significant differences in our reanalysis data (Fig.R2). The EN4 record shows the same variations and values over the analyzed period. The GLORYS record also reveals the same pattern between the two depths since 1993, even though the 50-400m SOTs are slightly colder by about 0.3°C, which might be due to the influence of cool Antarctic Surface waters. We therefore infer that we selected the right water depth (50-400m) in our study to monitor changes on mWDW intrusion on the continental shelf.

Figure R2. a) EN4 and b) GLORYS records at 50-400m (black) and 200-400m (blue) water depth.

Second, more discussion is needed for the Holocene TEX86L record, which is one of the main contributions of this study. Given the proximity of the study site to land, is there any significant overprint of terrigenous GDGTs on their TEX86L record? The authors also need to demonstrate that their result is insensitive to the choice of TEX86 index (TEX86, TEX86H or TEX86L). Why not use the more commonly applied TEX86 or TEX86H as this will aid comparison with other studies? Also important is to justify why it is reasonable to apply a 0-200m calibration to reconstruct the water temperatures at 50-400m. The temperature gradient at different water depths tend to differ, thus can result in different calibrations which in turn lead to different reconstructed absolute temperature values and also the amplitude of temperature change (e.g. using the SST calibration vs 0-200m). The choice of calibration can therefore affect the Holocene warming and the subsequent quantification of ice melt. Also, is there any evidence that the archaea at the study site mostly occur within this depth range? Can the authors propose some mechanisms by which the GDGTs below photic zone can be exported to the seafloor?

Response 8:

We now show the BIT index record (Fig.S2) calculated throughout the Holocene. It clearly appears that our record did not undergo any influence from terrestrial organic matter. The rapid warming from 8,200 to 7,000 and the following warming trend is therefore not impacted by terrigenous GDGTs. The TEX86 and TEX86H are used for temperate and warm temperatures, respectively. The TEX86L has been specifically developed for polar oceans and increasingly applied in the Southern Ocean (e.g. Kim et al., 2010; Kim et al., 2012; Etourneau et al., 2013; Sangiorgi et al., 2018). We therefore strongly believe that this calibration, despite a need of further improvements, is today the most suitable to estimate the Southern High Latitudes SOT. As mentioned in the manuscript, two studies from the Western Antarctic Peninsula (Murray et al., 1998; Kalanetra et al., 2009) showed that the archaea mostly grow within the 50-200m water depth. As demonstrated by Kim et al., 2012, the TEX86L reflects the temperature of the subsurface ocean (45-200m) and not the SST. Given the depth of the mWDW circulation, we suggest that the TEX86-GDGTs is strongly influenced by the shoaling of the wind-driven mWDW and therefore record its evolution through time. GDGTs are probably settling attached to several types of particles produced from the melting of sea ice or ice shelf along with fecal pellets or marine snow. Given the shallow depth of our study site (760m water depth), we suggest that the downward transport is probably relatively fast.

Third, given the strong seasonality in polar regions, is it possible that the GDGTs mainly reflect summer/winter instead of mean annual condition? Can seasonality explain the opposite trends in proxy-derived Holocene SA and SOT records, especially given the consistent trend in SAT and SOT derived from reanalysis products and climate models? If yes, then the main claim of the study that ocean plays a more dominant role than the atmosphere in Holocene ice shelf instability will no longer hold. Further, if proxies do record seasonal signal, how reasonable is it to apply mechanisms derived from mean annual reanalysis/observed data to explain the variations in the Holocene proxy data? The data analyses presented here are based on annual mean temperatures but I wonder if, summer or annual mean temperature, is more relevant for ice shelf instability as melting mostly occurs in summer?

Response 9:

The Traumarcheota synthesizing the GDGTs grow mostly during the late winter and early spring seasons but are produced throughout the year. (Murray et al., 1998; Kalanetra et al., 2009). While there are no studies focusing on that aspect particularly, the sediment integrates an annual mean signal that smoothes away the seasonality. We therefore suggest that the GDGT signal represents a smoothed annual

signal. Similarly, the isotope-based SAT also represent a smoothed annual signal (Stenni et al., 2017; Goursaud et al., 2018). In addition, Zitto et al., 2015 reported that seasonal SAT (spring, summer, autumn, winter) similarly evolved over the last century. Although orbital changes have modulated the seasonal insolation received by southern high latitudes at the Holocene timescale (Berger, 1978), the climate variations at the seasonal scale are lesser than the mean climate changes at the Holocene trend (Renssen et al., 2005; Pike et al., 2009).

Fourth, as it is, the link between SOT and ice-shelf instability is established based on the seemingly contemporaneous Larsen B collapse and SOT peak around year 2000. However, SOT was low during the collapse of other ice shelves, and the Ekman pumping anomalies hovered around 0 indicating neither upwelling nor downwelling was dominant. In other words, the mechanisms proposed by the authors to link SOT and ice-shelf instability do not seem to apply to most past ice shelf collapse events. Therefore, what is the likelihood of these processes occurring during the Holocene and in the future? And importantly, are they more likely than the processes that govern the collapse of all the other events?

Response 10:

After considering the Reviewer 1 suggestions, we deleted the EN4 reanalysis records prior to 1979, i.e. before the satellite era. This reinforces our hypothesis as the estimated SOT are significantly increasing between 1980 and 2000 along the EAP continental shelf during the major EAP collapses. Nevertheless, we specify in our discussion that both ocean and atmosphere have played a role together on these events as they share a similar profile.

Fifth, the paper needs more detailed description of the analyses performed on reanalysis data and climate models to help others replicate their work. For instance, how were the 0.3degC and 1.5degC Holocene warming calculated? For the SOT from the GLORYS and EN4: what is the definition of EAP margin - is it the same as the grey box in Figure 1? Why is it better to examine the SOT from the EAP margin instead of the SOT at the JPC38 site even though the latter will allow a better comparison with the Holocene TEX86L record? Calculation steps for the ice shelf melt rate need to be outlined so that others can reproduce the result. Please find more suggestions below in the form of specific comments.

Response 11:

We have added in the Method section a sentence explaining how we calculated the +0.3 and +1.5°C warming. In the Fig. 1 caption, we also mentioned where reanalyzed data and model simulations have been computed. Although the EAP margin is definitively larger and longer than the grey box shown in the Fig.1, we selected the area delineated by the continental shelf surrounding the EAP ice shelf. In the first sentence of Results and Discussion, we now referred to the Fig.1 so that the readers can immediately see what is considered as EAP margin in our study and where the reanalyses and models have been performed without any ambiguity. We examined the SOT in the entire shaded area as it corresponds to location where the wind-driven mWDW penetrates the shelf and flows towards the ice shelves (Larsen A, B and eventually C). We therefore consider that it is more suitable and accurate to scrutinize modern changes in a wider area than only at a specific site, given that the marine sediments also record a regional signal. We included in the Methods section the details regarding the calculation steps used to estimate the Ekman Pumping and the melting rate, and we also attached a file including the Ekman pumping code. Overall, we also provide many more details on the reanalysis data.

Specific comments

Line 42: Typo; "...Prince Gustave..." should be "...Prince Gustav..." Please correct this throughout the manuscript.

Response 12:

Corrected throughout the text and supplementary.

Line 80: "...from globally downwelling to upwelling conditions..." globally or regionally? The calculations are based on data from the EAP.

Response 13:

Corrected.

Line 87-90: I am not convinced by this argument. Stronger upwelling will bring more warm deep waters to intermediate depths, thus should increase the SOT within days - not years. On the other hand, the variability in SOT is much higher than that of SAT (Figure 2) in spite of a higher heat capacity in the ocean, whereas the SAT measured at Esperanza and Marambio stations show more variability which is more consistent with the SOT variability and Ekman pumping anomalies, including the peak at around year 2000. These rather curious results need to be further discussed.

Response 14:

Now is included in the Fig.1 the Esperanza and Marambio stations in order to show observation data instead of reanalysis ones. We also better explain the coherency between the different records in the revised version. As stated by reviewer 1 "the James Ross Island record is not directly air temperature but rather stable isotopes calibrated to recent air temperatures". The amplitude of the SAT reconstruction will change according to the ‰/T slope used. Mulvaney et al., 2012, used a slope of 0.8‰ °C⁻¹ while more recent studies argue that a slope of 0.4‰ °C⁻¹ (Stenni et al., 2017; Goursaud et al., 2018) should have been used, thus doubling the SAT variability and reconciling it with SOT variability.

Line 97-98: Not clear what is meant by "...a slight SOT increase in the context of atmospheric warming...". Please clarify.

Response 15:

Replaced by "in concert with atmospheric warming".

Line 128-131: TEX86L and TEX86H are based on different indices, thus can potentially result in different reconstructed temperature variations, including its trend (e.g. Taylor et al 2013) and variability (e.g. highly variable TEX86 record at ODP1098 of Shevenell et al 2010 vs. smoother TEX86L record at JPC-10 by Etourneau et al 2013). It is therefore important to demonstrate (at least in the Supplementary Information) that the reconstructed temporal trend at the study site is insensitive to the choice of the proxy index. Furthermore, the amplitude of reconstructed temperature change relies on the calibration used (due to the temperature range in the regression). Therefore, there needs to be justifications for using the 0-200m calibration to reconstruct temperature at 50-400m because (1) the temperature gradient at 50-400m (in a spatial temperature calibration dataset) is different than that at 0-200m, using a 0-200m calibration will therefore lead to erroneous temperature estimates including the amplitude of change; (2) any evidence that the sedimentary TEX86L at the study site originates from the water depths of 50-400m? In the calibration study of Kim et al (2012) it is stated that "...TEX86L predominantly reflects subsurface (the depth interval of ~45-200m)...". Also, what are the mechanisms that can transport the lipids from subsurface ocean below 200m to the

seafloor? (3) How does the TEX86L-inferred for the core-top compare to modern-day/late Holocene SOT?

Response 16:

See Responses 8 and 9.

Line 166: "constrains" should be "constraints".

Response 17:

Corrected.

Line 167-169: Unclear what are errors of a factor 2. Please rephrase.

Response 18:

Corrected.

Line 179-185: This argument is flawed. The calculation of the ice shelf melting is based on the assumption that SOT warming causes ice melting, therefore the result, ie. the calculated melt rate, cannot be presented as evidence for a dominant ocean forcing on regional ice shelf instability.

Response 19:

Although the calculation did not discard a role of the atmosphere, we only show that quantitative estimate are consistent with a large role of the ocean. In other words, the calculation shows that there is not only a correlation with SOT but that the heat available in the ocean is enough to melt large amount of ice. We now say "Therefore, although both ocean and atmosphere thermal forcing imparts on ice shelf instability, we conclude that, during the ocean warming phases, the SOT must have played a major role on controlling the regional ice shelf retreat". This comes in addition to previous sentences where it is clearly mentionned the important role of the atmosphere.

Line 193-195: Please show the simulation results from all the models in Supplementary Information as to rule out the possibility that the averaged trend is in fact dominated by a few models with relatively strong trends.

Response 20:

See Figure S4.

Line 210: Typo in "...due ton anthropogenic forcing..."; "ton" should be "to". I would add "projected" before "anthropogenic forcing" as those IPCC RCP scenarios are projections not reality.

Response 21:

Corrected.

Line 213: "though" should be "through".

Response 22:

Corrected.

Line 228-231: Given what's written here, I urge the authors to include the uncertainty of the ensemble mean (e.g. in the form of quantile or standard deviation) in Figure 2 so that

the reader can judge how reliable the reanalysis data are. In addition it will also be useful to add a Supplementary Figure showing all the realizations.

Response23:

We agree that uncertainty coming from observational data is an important caveat that we have now further discussed in the text. Since not all the datasets provide information about uncertainty, our choice to include systematically two different and independent observational dataset for both atmosphere (ERA-Interim and 20CR) and ocean (EN4 and GLORYS) was explicitly aimed at evidencing such uncertainty in the Antarctic region. Regarding the ocean temperature (Fig. 2c), only EN4 provides information about its uncertainty, which has been now added as an error bar in Fig.R3 below, while GLORYS does not provide such a field. Concerning the atmospheric reanalysis that we used to compute the Ekman pumping (Fig. 2b), the website allows to extract the mean wind field (and the associated error), but not the individual members used to calculate the mean wind field. From this single dataset, it is not possible to compute an error bar of the curl of the wind stress from which the Ekman pumping is derived from, since the curl is a difference of a gradient. To compute the uncertainty for the Ekman pumping we would thus need individual realisations, which are not available. Nevertheless, by comparing the different datasets in Fig.2, we further underline in the Method section of our manuscript that the uncertainty associated with the reanalysis is rather large in Antarctica due to very few data available to allow the data assimilation to correctly constrain the different fields. However, we consider them robust enough to estimate the reconstructed parameters of this study.

Figure R3. EN4 records and the associated ensemble mean, which clearly show that the variations reconstructed using the reanalysis data are statistically significant.

Line 232: "...to face SAT with Ekman pumping and SOT data." Unclear sentence, please rephrase.

Response 24:
Changed.

Line 233-235: More details on the calculation of Ekman pumping is needed in order to improve transparency and enable reproducibility in future studies. At the very least, the definition and unit should be stated. I also encourage the authors to share the code used to perform the data analyses.

Response 25:

The definition of Ekman pumping is $w_e = \frac{\text{curl}(\vec{\tau})}{\rho_0 f}$

where $\vec{\tau}$ is the wind stress vector, ρ_0 is the reference density of freshwater, f is the coriolis parameter.

The unit of Ekman pumping w_e is m/s but here it has been standardized by the standard deviation computed over the whole period of available data.

It has been added in the manuscript as well as the code used.

Line 237: The SOT were estimated from BOTH the GLORYS and EN4. Please rephrase to reflect this. I would also briefly mention why both datasets are needed for the SOT estimation (i.e. each covering different time intervals).

Response 26:
Modified.

Line 244 & 247: On what basis is the "initial temperature" for each reanalysis dataset determined?

Response 27:

We decided to use the absolute values instead of showing the anomalies in order to be consistent with the SAT based on measured data at the two meteorological Esperanza and Marambio stations.

Line 247-249: It is not very clear in the Supplementary Information how/why 50-400m is assumed. As shown in Figure S1, the thermocline at both sites extends down to 75m whereas the halocline extends further down to 200m. While the salinity profile at both sites do converge around 400m, the temperature profiles never do (the red site never approaches -1degC which characterizes the water mass).

Response 28:
See Response 7.

SI Line 8: "...flew..." should be "...flowed..."; "...Gustave..." should be "...Gustav..."

Response 29:
Corrected.

SI Line 15: "...near stations" should be "...nearby stations"

Response 30:

Corrected.

SI Line 16: Which previous studies? References needed here.

Response 31:

Deleted.

SI Line 44-45: Missing reference / further evidence for "...among the best datasets", why/how are they the best?

Response 32:

This paragraph has been removed as we now show the observation SAT data. The details regarding the Ekman Pumping and SOT have been moved to the Method section of the manuscript.

SI Line 47-50: To support your statement of "... share a high similarity" please provide statistics (e.g. correlation) or at least plot the reanalysis data next to the monitoring data in your Fig S2 to allow a better visual comparison. Also needed is justifications for opting reanalysis data over observed data at nearby sites (how near? provide coordinates of these sites or include them in one of the maps), especially in light of the structural similarity between the observed SAT time series and the SOT and Ekman pumping anomalies time series.

Response 33:

See Response 32.

SI Line 55-57: Robust enough or not depends on the context. Further clarification on how/why given the regional oceanography features it is reasonable to assume that the reanalysis data based on few observations are adequate to support the claims made by the authors.

Response 34:

See Response 32.

Figure S1: The authors need to clearly state the source of the hydrographic data used. Ocean Data View is a data visualization software not a database. It is also not clear if the data shown are annual mean or of certain month. Also needed here is a discussion on how well-constrained the hydrographic data are in this region and if the uncertainty may fundamentally change the working hypothesis (mWDW occurs between 50-400m at their study site).

Response 35:

See Response 7.

Figure S4 is missing.

Response 36:

Now are 4 figures are in the supplement.

References:

- Kilian, R. & Lamy, F. A review of Glacial and Holocene paleoclimate records from southernmost Patagonia (49-55°S). *Quater. Sci. Rev.* **53**, 1-23 (2012).
- Saunders, K.M. et al. Holocene dynamics of the Southern Hemisphere westerly winds and possible links to CO₂ outgassing. *Nat. Geosci.* **11**, 650-655 (2018).

- Zitto, M.E., Barrucand, M.G., Piotrkowski, R. & Canziani, P.O. 110 years of temperature observations at Orcadas Antarctic. *Roy. Meteor. Soc.* **36**, 809-823 (2015).
- Mayewski, P. et al. State of the Antarctic and Southern Ocean climate system. *Rev. Geophys.* **47**, RG1003, doi: 10.1029/2007RG000231 (2009).
- van Caspel, M., Schröder, M., Huhn, O. & Hellmer, H.H. Precursors of Antarctic Bottom Water formed on the continental shelf off Larsen Ice Shelf. *Deep-Sea Res. I* **99**, 1-9 (2015).
- Nicholls, K.W., Makinson, K. & Venables, E.J. Ocean circulation beneath Larsen C Ice Shelf, Antarctica from in situ observations. *Geophys. Res. Lett.* **39**, L19608 (2012).
- Camerlenghi, A. et al. Glacial morphology and post-glacial contourites in northern Prince Gustav Channel (NW Weddell Sea, Antarctica). *Mar. Geophys. Res.* **22**, 417-443 (2001).
- Herraiz-Borreguero, L. et al. Circulation of modified Circumpolar Deep Water and basal beneath the Amery Ice Shelf, East Antarctica. *J. Geophys. Res.: Oceans* **120**, 3098-3112 (2015).
- Jacobs, S. S., Hellmer, H., Doake, C.S.M., Jenkins, A. & Frolich, R. Melting of ice shelves and the mass balance of Antarctica. *J. Glaciol.* **38**, 375-387 (1992).
- Joughin, I. & Padman, L. Melting and freezing beneath Filchner-Ronne Ice Shelf, Antarctica. *Geophys. Res. Lett.* **30**, doi:10.1029/2003GL016941 (2003).
- Foldvik, A. et al. 2004. Ice shelf water overflow and bottom water formation in the Southern Weddell Sea. *J. Geophys. Res.* **109**, C02015, doi: 10.1029/2003JC002008 (2004).
- Liu, C., Wang, Z., Cheng, C., Xia, R., Li, B., Xie, Z. Modeling modified circumpolar Deep Water intrusions onto the Prydz Bay continental shelf, East Antarctica. *J. Geophys. Res. Oceans* **122**, 5198-5217 (2017).
- Nitsche, F.O. et al. Bathymetric control of warm ocean water access along the East Antarctic Margin. *Geophys. Res. Lett.* **44**, 8693-8944 (2017).
- Kim, J.H. et al. New indices and calibrations derived from the distribution of crenarchaeal isoprenoid tetraether lipids: Implications for past sea surface temperature reconstructions. *Geochim. Cosmochim. Acta* **74**, 4639-4654 (2010).
- Kim, J.-H., et al. Increase in Late Holocene subsurface temperature variability in East Antarctica. *Geophys. Res. Lett.* **39**, L06705 (2012).
- Etourneau, J. et al. Holocene climate variations in the western Antarctic Peninsula: evidence for sea ice extent predominantly controlled by changes in insolation and ENSO variability. *Clim. Past* **9**, 1431-1446 (2013).
- Sangiorgi, F. et al. Southern Ocean warming and Wilkes Land ice sheet retreat during the mid-Miocene. *Nat. Comm.* **9**, doi: 10.1038/s41467-017-02609-7 (2018).
- Murray, A. E. et al. Seasonal and spatial variability of bacterial and archaeal assemblages in the coastal waters near Anvers Island, Antarctica. *Appl. Environ. Microbiol.* **64**, 2585-2595, 1998.
- Kalanetra, K. M., Bano, N. & Hollibaugh, J. T. Ammonia-oxidizing Archaea in the Arctic Ocean and Antarctic coastal waters. *Environ. Microbiol.* **11**, 2434-2445 (2009).
- Stenni, B. et al. Antarctic climate variability at regional and continental scales over the last 2,000 years. *Clim. Past* **13**, 1609-1634 (2017).
- Goursaud, S., Masson-Delmotte, V., Favier, V., Orsi, A. & Werner M. Water stable isotope spatio-temporal variability in Antarctica in 1960-2013: observations and simulations from the ECHAM5-wiso atmospheric general circulation model. *Clim. Past* **14**, 923-946 (2018).
- Berger, A.L. Long-term variations of daily insolation and Quaternary climatic changes. *Am. Meteor. Soc.* **35**, 2362-2367 (1978).
- Renssen, H., Goosse, H., Fichefet, T., Masson-Delmotte, V. & Koç, N. Holocene climate evolution in the high-latitude Southern Hemisphere simulated by a coupled atmosphere-sea-ice-ocean-vegetation model. *Holocene* **15**, 951-964 (2005).
- Pike, J. et al. Observations on the relationship between the Antarctic coastal diatoms *Thalassiosira antarctica* Comber and *Porosira glacialis* (Grunow) Jorgensen and sea ice concentrations during the Late Quaternary. *Mar. Micropal.* **73**, 14-25 (2009).

Reviewer #1 (Remarks to the Author):

I am happy to recommend publication in Nature Communications as all my queries/critiques have been satisfactorily addressed.

Reviewer #3 (Remarks to the Author):

The paper of Etourneau et al. is a revised version of manuscript NCOMMS-18-15836 that I previously reviewed. I find the revised manuscript much improved and have just a few concerns and suggestions that I would like the authors to address before the publication of the manuscript.

I agree that each proxies has its pros and cons, and that it is beyond the scope of this paper to improve the proxy calibration used to convert proxy values to temperature. However, although the authors strongly believe that the TEX86L is more suitable for the application in the polar regions, as far as I can tell there is yet a consensus on this matter among the community. In this regard, I would urge the authors to show in the Supplementary Information (SI) a comparison of the commonly used TEX86 vs. their chosen index TEX86L. In light of several studies (see my comments on the previous version of the manuscript) that reported different temporal trends and values resulted from these two indices, such a comparison will help support the authors' choice of proxy index and the claim made in the Supplementary Information Line 61-62.

Some readers from the paleoclimate community (especially non-users of TEX86) might find it confusing that the authors used a 0-200m calibration to reconstruct water temperatures at 50-400m, in spite of the cited studies which suggest that the GDGT producers dwell in the subsurface water depths of 45-200m ie excluding the sea surface. Ideally, the solution would be to recalibrate the proxy index using the water temperature of 50-400m, which will surely result in a different calibration slope than the 0-200m calibration applied, and thus directly affects one of the main findings of the paper, namely the reconstructed amplitude of temperature change, i.e., the 0.3 - 1.5degC which are then used to calculate the ice melt rate. A less ideal solution would be to include Figure R2 (from the rebuttal letter, which shows similar temperature evolution at 50-200m and 200-400m in the study area) in the SI to support the authors' argumentation on why it is possible to use reconstructed 0-200m temperature record to approximate the temperature evolution of 50-400m at their study site.

Specific comments

Line 44-45: "... eventually resulting in 2017 into a massive iceberg..." sounds awkward. A giant crack cannot result in the year 2017. Please rephrase.

Line 76-80: Long and confusing. Please rephrase.

Line 91-92: I do not think this is physically possible. The temperature increase should occur as soon as the warmer subsurface ocean water is upwelled. Ocean heat capacity would however act to slow down the temperature change in the upwelled water that is now exposed to the much colder atmosphere / surrounding water mass.

Line 100-101: "The concomitance of the rapid SOT warming around the EAP ice shelf with a series of regional ice shelf collapses, such as that of the Larsen B ice shelf," This is only true for the Larsen B ice shelf. The collapse of other ice shelves ie Larsen A, Prince Gustav and Larsen Inlet do not coincide with the warming. I would rephrase this.

Line 103-105: Not sure if this helps to convince the reader of the link between the ice shelf collapse and the upwelling/downwelling regime and/or SOTs because multiple ice shelf collapses occurred around 1990s, a time period that is characterized by similar SOTs and a downwelling regime as in the early 2000s where the authors argued to be a period without any known ice shelf collapse.

Given the comments on Line 100-101 and 103-105, I think it might help clarify the link between ice shelf collapse and ocean/atmosphere forcing if the authors rephrase this paragraph to emphasize/focus on the Larsen B event?

Line 121: "evidences" As far as I know, the noun "evidence" is uncountable, just like knowledge and information.

Line 149-150: There is only ONE paleorecord in Figure 3c. Please rephrase the sentence to reflect this, or include more paleorecords in Figure 3.

Line 204-205: "Replacing these results into an Holocene context" sounds odd. Perhaps "Placing these results in a Holocene context"?

Line 263: "converting TEX86L values into sea surface temperatures" Shouldn't it be subsurface ocean temperatures?

Figure 1: "Prince Gustave Channel" should be Prince Gustav Channel.

Figure 2: It is difficult to distinguish the dark blue curve vs the black curve in panel a and the light blue curve vs dark blue curve. Using contrasting colors for the curves of the same panels might help. Also, there are two y-axes in panel a; it is not clear from the figure to which axis the temperature records correspond.

Supplementary Information

Line 14-16: Reference for the Southern Ocean Atlas data is missing. Also there is a typo "Gustave".

Response to reviewers

We would like first of all to thank the two reviewers for positive recommendations and suggestions. Here we address a response to Reviewer 3 only as the Reviewer 1 suggests publication.

Reviewer #3:

The paper of Etourneau et al. is a revised version of manuscript NCOMMS-18-15836 that I previously reviewed. I find the revised manuscript much improved and have just a few concerns and suggestions that I would like the authors to address before the publication of the manuscript.

I agree that each proxies has its pros and cons, and that it is beyond the scope of this paper to improve the proxy calibration used to convert proxy values to temperature. However, although the authors strongly believe that the TEX86L is more suitable for the application in the polar regions, as far as I can tell there is yet a consensus on this matter among the community. In this regard, I would urge the authors to show in the Supplementary Information (SI) a comparison of the commonly used TEX86 vs. their chosen index TEX86L. In light of several studies (see my comments on the previous version of the manuscript) that reported different temporal trends and values resulted from these two indices, such a comparison will help support the authors' choice of proxy index and the claim made in the Supplementary Information Line 61-62.

Some readers from the paleoclimate community (especially non-users of TEX86) might find it confusing that the authors used a 0-200m calibration to reconstruct water temperatures at 50-400m, in spite of the cited studies which suggest that the GDGT producers dwell in the subsurface water depths of 45-200m ie excluding the sea surface. Ideally, the solution would be to recalibrate the proxy index using the water temperature of 50-400m, which will surely result in a different calibration slope than the 0-200m calibration applied, and thus directly affects one of the main findings of the paper, namely the reconstructed amplitude of temperature change, i.e., the 0.3 - 1.5degC which are then used to calculate the ice melt rate. A less ideal solution would be to include Figure R2 (from the rebuttal letter, which shows similar temperature evolution at 50-200m and 200-400m in the study area) in the SI to support the authors' argumentation on why it is possible to use reconstructed 0-200m temperature record to approximate the temperature evolution of 50-400m at their study site.

The calibration of the TEX86 in polar regions is a critical issue since this proxy has been increasingly used in paleoclimate studies to reconstruct ocean temperatures. While the calibration can be obviously refined in Antarctic marine environments in order to get a better estimate of the reconstructed temperatures, it remains difficult and a long-standing work due to the limited access to adequate samples, especially in coastal and sea ice-covered areas. We already referred to several recent studies dealing with this issue in our former rebuttal letter. Among many others, we could cite the work of Ho et al. (2014) who clearly stated that “a regional TEX86L calibration is more suitable equation in polar and subpolar regions for temperature reconstruction than the global calibration.” Indeed, the TEX86 calibration does show a plateau in estimated temperatures below 5°C preventing its use in polar regions (Kim et al., 2010). The TEX86L alleviates this plateau and is therefore much more suitable for the Southern Ocean.

More specifically, when applying the TEX86 calibration to our data (see figure R1), as suggested by the reviewer, the converted temperatures using two global calibration (green and red) reveal values higher than +10°C on average, some reaching up to almost +16°C and a warming event at 8.2 kyrs of +8°C. These unrealistic and warm values cannot be found in any coastal Antarctic area, but at best in the Subantarctic zone, if not further north. This calibration, in addition to estimate elevated values, also slightly modifies the general trend

and amplitude, as suspected by the reviewer her/himself. We can provide such figure in our supplementary information if required; however, we really question its relevance as it may confuse the readers and as stated by the reviewer the calibration issues are beyond the scope of this paper.

In addition, although the calibration has been developed for the 0-200m water depth, we here show that there are no significant differences in terms of trend and values with other sub-surface water depth used for this study at the core site and EAP. More precisely, EN4 reanalysis shows no difference in temperatures between the 0-200m, 50-400m and 200-400m depths in term of absolute values and inter-annual variability while GLORYS reanalysis suggests a $\sim 0.2^{\circ}\text{C}$ difference in temperatures between the 50-400m and 200-400m and a $\sim 0.8^{\circ}\text{C}$ difference between the 0-200m and the other depth range. The inter-annual variability is however preserved. We suggest that the 0-200 m cooler temperature results of the autumn-winter cooling of the top meters of the water column. Taking these results at face value may indicate that our reconstructed SOT in terms of values and amplitude might be slightly underestimated. However, it is clear that these small differences and the fact that inter-annual variability is preserved whatever the depth we look at in the reanalyses do not affect the interpretations, especially as the calibration is performed at the Southern Ocean scale (Kim et al., 2010). At this large scale, annual SST, 0-200 m and 50-400m annual SOT are strongly correlated due to convection processes during the winter (WOA, 2013). To avoid any confusions and to reinforce our arguments that the use of the TEX86L calibration is the best for the 50-400m layer, we have decided to include an improved version of the Fig.R2 and further develop this point in our SI (lines 62-76).

Finally, our data seem to be well supported by model simulations. During the first stage of our submission, Reviewer 2 stated in his comments: "It struck me immediately that this early-mid-Holocene increase is also seen in transient climate model simulations - especially the TRACE21K simulation from Feng He. I've attached a plot of zonally averaged potential temperature at 400m depth for the region 60-79S. It has more than a passing resemblance to the panels in Fig 3. So if the authors wanted to strengthen their argument, it might be worth including this kind of model output?" Comparing our data with model simulations will be another task performed in the frame of another article. However, it proves again that using our TEX86L calibration is the best way to reconstruct SOT along the EAP.

Figure R1. Different calibrations used to convert the TEX86 into sea subsurface temperatures at the JPC-38 core site: TEX86L (blue) as shown in our study (kim et al., 2010), TEX86 based on Schouten's calibration (Green) (Schouten et al., 2002) and Shevenell's one (red) (Shevenell et al., 2011).

Specific comments

Line 44-45: "... eventually resulting in 2017 into a massive iceberg..." sounds awkward. A giant crack cannot result in the year 2017. Please rephrase.

We rephrased this sentence according to the reviewer's comment: "Since 2010, a giant crack has continuously incised the Larsen C ice shelf until it broke off in 2017 to form a massive iceberg of ~6,000 km² (~9-12% of the total ice shelf)⁵, thus drawing the premise of unprecedented major collapses in the near future."

Line 76-80: Long and confusing. Please rephrase.

To make it clearer, we first shortened the sentence in the main text: "This dominant downwelling regime over our study area probably results of a northern position of the convergence zone between the southern westerly winds (SWW) and the southerly winds, blowing northward along the south EAP (See Supplementary Fig. 3)." We moved some of the explanations in the captions of Fig.S3: "Black arrows indicate the major wind direction and strength, i.e. the mean position of the southern westerly winds (SWW) favoring upwelling, and the southern winds, south of 66°30'S, conducive to mean downwelling conditions."

Line 91-92: I do not think this is physically possible. The temperature increase should occur as soon as the warmer subsurface ocean water is upwelled. Ocean heat capacity would however act to slow down the temperature change in the upwelled water that is now exposed to the much colder atmosphere / surrounding water mass.

We have included the reviewer's suggestions: "The temperature increase should quickly respond to the upwelled warmer subsurface ocean water. However, ocean heat capacity could act to slow down the temperature change in the upwelled water that is exposed to the much colder atmosphere and surrounding water mass."

Line 100-101: "The concomitance of the rapid SOT warming around the EAP ice shelf with a series of regional ice shelf collapses, such as that of the Larsen B ice shelf," This is only true for the Larsen B ice shelf. The collapse of other ice shelves ie Larsen A, Prince Gustav and Larsen Inlet do not coincide with the warming. I would rephrase this.

Line 103-105: Not sure if this helps to convince the reader of the link between the ice shelf collapse and the upwelling/downwelling regime and/or SOTs because multiple ice shelf collapses occurred around 1990s, a time period that is characterized by similar SOTs and a downwelling regime as in the early 2000s where the authors argued to be a period without any known ice shelf collapse.

Given the comments on Line 100-101 and 103-105, I think it might help clarify the link between ice shelf collapse and ocean/atmosphere forcing if the authors rephrase this paragraph to emphasize/focus on the Larsen B event?

We slightly modified these two sentences to follow the reviewer's suggestion: "We also find that the most rapid and abrupt SOT warming reconstructed around the EAP ice shelf over the last centuries concurred with the Larsen B ice shelf collapse, thus strongly suggesting that a slight SOT increase, in concert with atmospheric warming, may have been pivotal in controlling the Larsen B ice shelf instability. While we miss robust observation data prior the 1970's, the continuous SOT increase during phases of smaller collapses could potentially imply that SOTs may also have contributed to other smaller ice shelf collapses along the EAP and around Antarctica."

Line 121: "evidences" As far as I know, the noun "evidence" is uncountable, just like knowledge and information.

Corrected.

Line 149-150: There is only ONE paleorecord in Figure 3c. Please rephrase the sentence to reflect this, or include more paleorecords in Figure 3.

We corrected the sentence as follow: "Several studies reported an intensification and southward migration of SWW during the same period of time⁴²⁻⁴⁴, as illustrated for instance by pollen records in Patagonia⁴², South America (Fig. 3c)."

Line 204-205: "Replacing these results into an Holocene context" sounds odd. Perhaps "Placing these results in a Holocene context"?

Corrected following the reviewer suggestion.

Line 263: "converting TEX86L values into sea surface temperatures" Shouldn't it be subsurface ocean temperatures?

It is true. Corrected.

Figure 1: "Prince Gustave Channel" should be Prince Gustav Channel.

Corrected

Figure 2: It is difficult to distinguish the dark blue curve vs the black curve in panel a and the light blue curve vs dark blue curve. Using contrasting colors for the curves of the same panels

might help. Also, there are two y-axes in panel a; it is not clear from the figure to which axis the temperature records correspond.

We now use more contrasted colors for each records shown in Fig. 2 and added the name of each y-axis corresponding to the Esperanza and Marambio station records.

Supplementary Information

Line 14-16: Reference for the Southern Ocean Atlas data is missing. Also there is a typo "Gustave".

We added the reference (S2) and corrected Gustave by Gustav in the text.